# SHINe: Simulator for Satellite on-Board High-Speed Networks Featuring SpaceFibre and SpaceWire Protocols

**Alessandro Leoni** [1,*]**, Pietro Nannipieri** [1]**, Daniele Davalle** [2] **and Luca Fanucci** [1]
**and David Jameux** [3]

1    Department of Information Engineering, University of Pisa, 56100 Pisa, Italy;
     pietro.nannipieri@ing.unipi.it (P.N.); luca.fanucci@unipi.it (L.F.)
2    IngeniArs s.r.l., 5610 Pisa0, Italy; daniele.davalle@ingeniars.com
3    Space Research and Technology Centre, European Space Agency, 2201 AZ Noordwijk, The Netherlands;
     david.jameux@esa.int
*    Correspondence: alessandro.leoni@ing.unipi.it; Tel.: +39-05027660

**Abstract:** The continuous innovation of satellite payloads is leading to an increasing demand of data-rate for on-board satellite networks. In particular, modern optical detectors generate and need to transfer data at more than 1 Gbps, a speed that cannot be satisfied with standardized technologies such as SpaceWire. To fill this gap, the European Space Agency (ESA) is supporting the development of a new high-speed link standard, SpaceFibre. SpaceFibre provides a data-rate higher than 6.25 Gbps, together with the possibility to use multiple Virtual Channels running over the same physical link, each one configurable with flexible Quality of Service parameters. These features make a SpaceFibre network very appealing but also complex to set up in order to achieve the desired end-to-end requirements. To help this process, a Simulator for HIgh-speed Network (SHINe) based on the open-source toolkit OMNeT++ has been developed and is presented in this paper. It supports the simulation of SpaceFibre and SpaceWire protocols in order to help both the final steps of the standardization process and the system engineers in the setup and test of new networks. SHINe allows to precisely simulate common network metrics, such as latency and bandwidth usage, and it can be connected to real hardware in a Hardware-in-the-Loop configuration.

**Keywords:** SpaceFibre; SpaceWire; network simulator; on-board satellite networks; OMNeT++

## 1. Introduction

Science and earth observation missions are experiencing a constant technology evolution, with their payload instruments needing higher and higher data-rates in order to stream the generated data. Typical examples are high-resolution optical payloads, as demonstrated in missions like MTG [1], Juice [2] and Plato [3]. In addition to the increased demand of data-rate, there is also a need to reduce satellite complexity by reducing the number of different on-board network technologies, and hopefully using one single network for all kinds of traffic in the future. This would bring a great harness reduction, together with a noticeable simplification of the overall system management. SpaceFibre [4] is the solution, supported by the European Space Agency (ESA) [5], aiming at solving all these problems. It supports a link speed up to 6.25 Gbps per lane, with the possibility to run up to 16 lanes in parallel, on both copper and optical fibre. Moreover, SpaceFibre provides up to 32 Virtual Channels (VCs) per link, each one independently configurable with different Quality of Service parameters (priority level, reserved bandwidth, and assigned time-slots). The use of Virtual Channels allows carrying different traffic classes using the same network technology. Hence, the network infrastructure



can be effectively shared among different applications, even when the requirements are completely different. In particular, it would be possible to use a SpaceFibre network both for payload applications (usually requiring high data-rates but with little time requirements) and for platform applications (usually requiring low data-rates but with stringent time and reliability requirements). These features represent a big step forward in comparison with older technologies such as SpaceWire [6,7] and make the configuration of a SpaceFibre network undoubtedly complex. Moreover, The core layers of the SpaceFibre standard have already underwent public review and are expected to be released as a standard in 2019, while the SpaceFibre network layer is still under standardization process and some features have yet to be defined. In order to help to finalize the standard and to simplify the development of new SpaceFibre networks, the Simulator for HIgh-speed Networks (SHINe) has been developed [8]. SHINe is a discrete event simulator supporting both SpaceFibre and SpaceWire protocols and it is entirely based on the open-source framework OMNeT++ [9]. With SHINe, it is possible to easily deploy a network via drag&drop from a palette and simulate it, collecting and analysing the results. Being based on OMNeT++, SHINe is completely written in C++ and it is easily extensible, allowing the user to develop custom nodes to use together with the existing ones. SHINe implements not only the basic SpaceFibre and SpaceWire protocols, but it also offers a Routing Switch node, both Remote Memory Access Protocol (RMAP) Target and RMAP Initiator nodes, and an advanced Hardware-in-the-Loop mechanism to connect physical devices to the simulator.

After this introduction, Section 2 presents an overview of the already existing network simulators in this field. Section 3 provides internal details of the SHINe software architecture. Section 4 shows an example of a simple network setup, briefly going through all the steps needed to define and simulate a network. Finally, the conclusions are drawn in Section 5.

## 2. Related Works

There are other simulators, presented in Table 1, already available for the simulation of SpaceWire and SpaceFibre networks. Among them, Modelling of SpaceWire Traffic (MOST) [10], developed by Thales Alenia Space (TAS), is probably the most used. MOST is based on the toolkit OPNET, hence it requires an annual license to be used, while OMNeT++ requires a license only for commercial applications and it is free for academic and research projects. Thanks to its long-term development, MOST provides a wide library of SpaceWire-related products, such as the SMCS116SpW, the SMCS332SpW, the Remote Terminal Controller (RTC) and the SpW-10X Switch. It also supports Spacefibre endpoints in its OPNET version. In comparison, SHINe provides only ideal components. Currently, TAS is porting MOST from OPNET to NS-3, which does not require the payment of a license. It appears that support on SpaceFibre has been recently added [11], however, no detailed public information are available. Another important simulator is SANDS [12], developed by Saint-Petersburg University of Aerospace Instrumentation (SUAI). SANDS aims to support the topology design for SpaceWire networks taking into account several parameters, such as the required fault-tolerance level, total network mass including cables and nodes, and power consumption. SANDS also allows simulating the network, using its SystemC engine, either at bit level or at packet level, as well as providing support for the generation of the scheduling tables for scheduling Quality of Service of STP-ISS [13] transport protocol running on top of SpaceWire.

While each simulator has its own strength, SHINe is the only one supporting both SpaceFibre and SpaceWire that is free, making it a very good product to help the development of such protocols.

**Table 1.** Summary of the main SpaceWire/SpaceFibre network simulators available.

| Simulator | Toolkit/Language | SpaceWire/SpaceFibre Support |
| --- | --- | --- |
| SHINe | OMNeT++ (C++) | SpW/SpFi |
| MOST (NS-3) | NS-3 (C++) | SpW/SpFi |
| MOST (OPNET) | OPNET (C) | SpW/SpFi |
| SANDS | Own/SystemC | SpW |

## 3. SHINe Core Building Blocks

This section provides detailed information on the software architecture of SHINe. After a short overview of the main mechanisms behind the simulator, the core building blocks are illustrated.

### 3.1. SHINe Software Architecture

The driving idea during the development of SHINe was to create a tool providing the building blocks to easily set up a SpaceFibre or SpaceWire network infrastructure, while the definition of the applications connected to this network is left up to the user. With this goal in mind, the main effort has been spent simplifying the Application-to-Network interface, hence most of the protocol-related metrics are observable and automatically recorded (packet latency, link usage, Flow Control Token (FCT) credit, generated packet size, received packet size). The main two core building blocks provided by SHINe are the SpaceFibre Endpoint and the SpaceWire Endpoint. They offer a C++ interface compliant with the service interface described in the two standards, completely hiding the protocols details to the application. Internally, both the SpaceFibre Endpoint and the SpaceWire Endpoint modules are composed by several nested submodules (Port and Codec), each one reducing the abstraction level down to the actual C++ implementation of the standard specifications. However, the application can treat the Endpoints as black boxes and it is not necessary for the user to know the details of the SpaceFibre and the SpaceWire standards.

As shown in Figure 1, the user-defined Application must extend a specific C++ interface in order to be connected to the Endpoints. In such a way, it is possible for the Endpoint to call "callback" functions of the application without knowing their actual implementation. For example, the Endpoint may notify through a callback that new data is ready to be read or that the transmission buffer has a free slot to send new data.

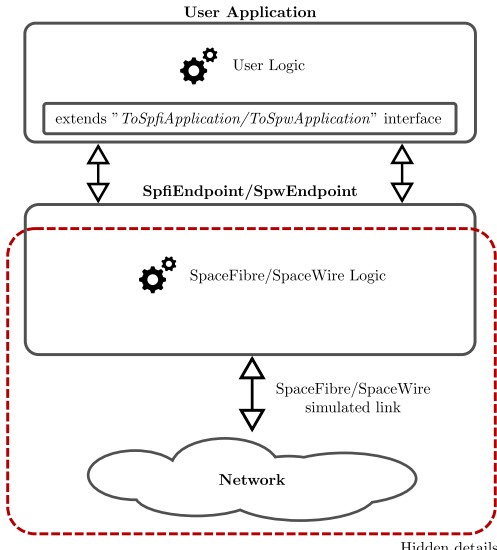

**Figure 1.** Interaction between User Application and SpaceFibre or SpaceWire Endpoint.

### 3.2. SpaceFibre Endpoint

The SpaceFibre Endpoint module is the core building block in SHINe to instantiate a fully-functional SpaceFibre port. As shown in Figure 2, it provides two input/output gates for the connection with the upper layer application and one input/output gate representing the physical SpaceFibre connector. Note that there are two gates towards the upper layer, one for the NChar transmission (*appPacket*) and one for the Broadcast messages transmission (*appBroadcast*), allowing using two different applications in case the user prefers to model the two interfaces separately. In order to be connected to a SpaceFibre Endpoint, an application must extend the *ToSpfiApplication* C++ abstract class. This class represents the interface that the Endpoint can use to notify about the availability of new NChars or Broadcast

messages to read. A SpaceFibre Endpoint automatically collects several metrics about network usage, such as: (i) packet latency; (ii) packet inter-arrival time; (iii) packet inter-transmission time; and (iv) broadcast message latency.

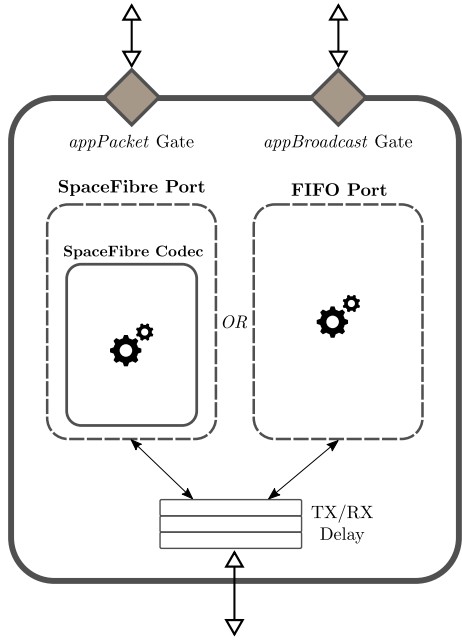

**Figure 2.** Internal architecture of a SpaceFibre Endpoint.

The SpaceFibre Endpoint is a versatile module and it can include either a SpaceFibre Port or a First In First Out (FIFO) Port, as specified in the standard, depending on a user defined parameter. The FIFO Port does not implement any specific transfer protocol and it is only capable of transferring NChars between the two far-ends, simulating a generic FIFO-like protocol but offering the same interface to the application of a SpaceFibre Port. Note that it does not provide a way to transmit Broadcast messages as well as a flow control mechanism, so data may be lost. When the SpaceFibre Port is chosen, the whole SpaceFibre protocol as described in the standard is used. In particular, the following features are supported:

- Fault Detection Isolation and Recovery (FDIR) mechanism, with retransmission in case of (injected) errors;
- Flow Control Mechanism (FCT);
- IDLE words, IDLE Frames and SKIP words insertion (See [4] for details on SpFi control words);
- Multilane layer support;
- Upper lane layer support, excluding 8b/10b encoding and serialisation. The OMNeT++ messages exchanged between SpaceFibre Endpoints represent an abstraction of the 40-bits SpaceFibre words. The choice to simulate at "word level" instead of at "bit level" has been taken because, from a networking standpoint, the simulation of the physical layer does not add any additional value to the results but it greatly affects the simulation time;

Both the Lane and the Multilane Layers can be bypassed to save additional simulation time if the user is not interested in simulating their behaviour and the overhead they add to the protocol. Each deployed SpaceFibre Endpoint can be independently configured through all the parameters foreseen by the standard, such as the Expected Bandwidth, the Priority levels, the Assigned Timeslots, etc., plus additional parameters such as buffers size and number of lanes. It is also possible to define a transmission and reception delay at the interface with the link to simulate internal buffers or pipeline stages in the data-path.

*3.3. SpaceWire Endpoint*

The SpaceWire Endpoint plays a role similar to the SpaceFibre Endpoint and it is the building block providing a fully functional implementation of the SpaceWire standard. From the user perspective, the two kinds of Endpoints offer a similar interface. A SpaceWire Endpoint, whose architecture is shown in Figure 3, provides two input/output gates for the connection with the upper layer application and one input/output gate representing the physical connector. The connection with the application is split in two gates, one for the NChars transmission and reception and one for the TimeCodes, hence it is possible to use two different applications for the packet stream and the TimeCodes. Note that Interrupts are not currently supported in SHINe. The SpaceWire Endpoint internally instantiate a TimeCode Manager, which is responsible for:

- keeping trace of the value of the next TimeCode to send;
- storing the value of the latest TimeCode received, in order to be able to validate (or discard) the next one. As specified in the standard, a TimeCode is valid when its value is equal to the value of the previous one plus one. Only if this condition is satisfied the application is notified about the reception of a new TimeCode.

Following the same logic of the SpaceFibre Endpoint, an application must extend the *ToSpwApplication* C++ abstract class in order to be connected to a SpaceWire Endpoint. In addition, in this case, several network related statistics are automatically collected by SHINe.

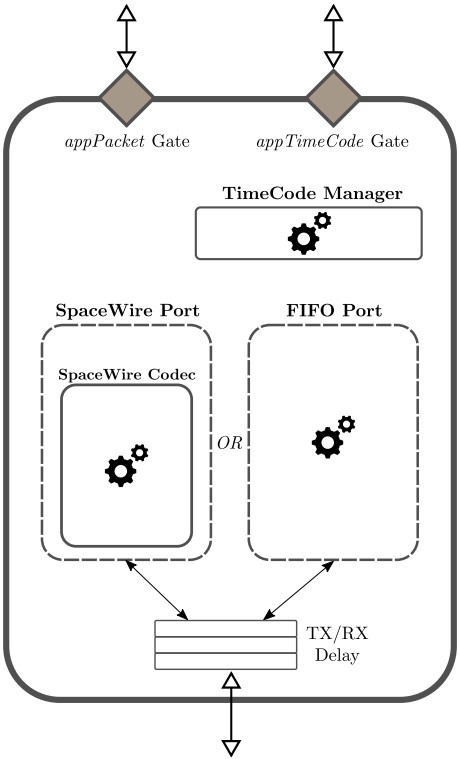

**Figure 3.** Internal architecture of a SpaceWire Endpoint.

A SpaceWire Endpoint can instantiate either a SpaceWire Port or a FIFO Port, keeping the same interface towards the application. In the former case, a complete SpaceWire Codec is simulated in C++, while in the latter case a generic FIFO-like protocol takes place, with the same limitations already described for SpaceFibre. When the SpaceWire Port is used, the most remarkable functions are:

- Simulation of the Initialisation phase, according to the initialisation state machine described in the standard;
- Flow Control Mechanism (FCTs);

- Error detection, in case a character is received when not expected or it contains an invalid field;
- ESC (see [6] for details) character insertion when no data is available to be sent;

Two adjacent SpaceWire Endpoints communicate exchanging OMNeT++ messages representing the characters sent by the Codecs. Note that, differently from SpaceFibre, in SpaceWire the different kinds of characters have different bit sizes. For example, an NChar is 10 bits long considering the Parity and Control bits, while an FCT is only 4 bits long, hence they take a different amount of time to be transmitted. The Endpoint is highly customizable through several parameters, such as transmission and reception data-path delay, initialisation link speed, buffers size, and others.

### 3.4. Routing Switch

The core elements in a network are naturally the Routing Switches. One of the most critical points in the study and development of the SpaceFibre Network Layer is its interoperability with legacy SpaceWire networks. A realistic scenario in the transition phase between the two technologies foresees the use of SpaceFibre as a high-speed backbone with some of the nodes connected through SpaceFibre and some others through SpaceWire. While the two protocols are compatible from the packet format standpoint, there are no specifications about:

- how to bridge Broadcast messages with the TimeCodes;
- how to forward packets between the two domains, where in SpaceFibre a packet is always associated to its Virtual Channel while in SpaceWire it is not;

The Routing Switch available in SHINe allows connecting both SpaceFibre and SpaceWire Endpoints and implements realistic solutions to the open points listed above.

An overview of the Routing Switch architecture is shown in Figure 4. A Routing Switch comprises a user defined number of ports of three possible kinds: (i) SpaceFibre Ports; (ii) SpaceWire Adapter Ports; or (iii) FIFO Ports. All of them share the same SpaceFibre interface to the internal Switching Matrix, but they differ in their implementations. In particular, the SpaceWire Adapter Port acts as a wrapper around a normal SpaceWire Port, making it to look like a SpaceFibre Port with only one Virtual Channel. Moreover, this wrapper is responsible for the bridging of Broadcast messages according to the following rule: whenever a TimeCode is received from the underlying SpaceWire port, it is wrapped into a Broadcast message of a specific type and then forwarded to the Switching Matrix to be propagated. Vice-versa, whenever the Switching Matrix tries to propagate a Broadcast message to a SpaceWire Adapter Port, the message is checked for its type: if it contains a TimeCode, the message is unwrapped and the TimeCode is sent, otherwise it is simply dropped. This simple mechanism makes possible for a SpaceFibre backbone to transparently carry TimeCodes. Moreover, the automatic loop prevention foreseen by the SpaceFibre standard guarantees that no multiple copies of the same TimeCode are propagated.

The SHINe Routing Switch implements all the features required by the SpaceFibre standards. In particular, it supports:

- Path and Logical Addressing. When using Path Addressing, the output port of the Routing Switch is directly written in the packet header. When using Logical Addressing, however, a Routing Table is necessary to associate the logical addresses to the output ports. The table can be provided by the user as a comma-separated file or can be built automatically by SHINe using Dijkstra algorithm in order to speed up the setup of the network;
- Virtual Network Mapping. For each Virtual Channel of each port, the user can specify the Virtual Network it belongs to. According to the SpaceFibre standard, packets can flow only through Virtual Channels belonging to the same Virtual Network. Note that, thanks to the Routing Switch implementation in SHINe, the single Virtual Channel of a SpaceWire Adapter Ports can be mapped into any Virtual Network.
- Multicast Support. In case the user decides to specify a custom Routing Table, he can define, for a specific logical address, a Multicast set of output ports. When Multicast is used, an NChar is

transferred from the input to all the output ports of the set at the same time, provided that they can all accept it.

- Group Adaptive Routing Support. Similarly to Multicast, a custom Routing Table can include a set of output ports to be used for Group Adaptive Routing. When an input port requires a new wormhole to be open, only the first output port in the set that is found free is chosen, improving network performances.

As already mentioned, the Routing Switch completely supports the Broadcast message mechanism, including the loop prevention and the per-channel timeout. In conclusion, the Routing Switch provided in SHINe allows to build mixed SpaceWire and SpaceFibre networks in order to test both network performances and protocols' interoperability.

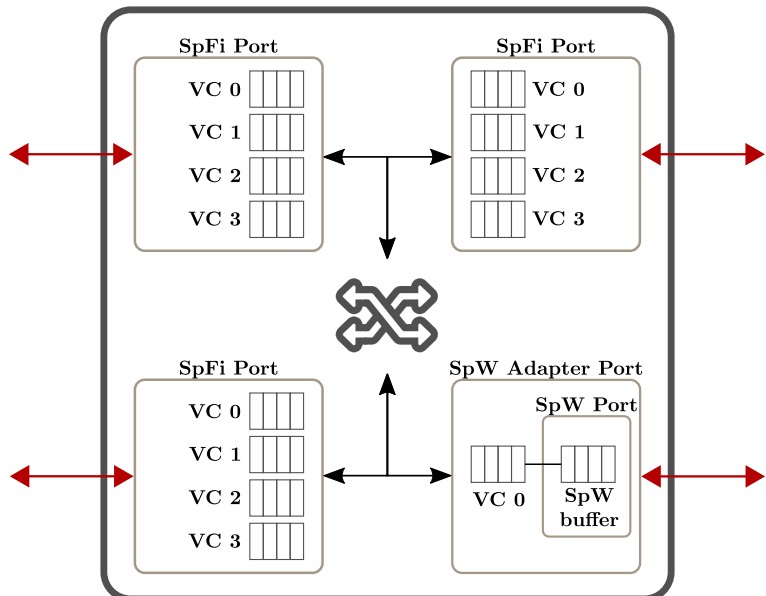

**Figure 4.** Routing Switch with a SpaceWire Adapter Port.

## 4. SHINe Additional Building Blocks

In this section, some additional modules available in SHINe are presented. They are not directly necessary to build a SpaceFibre or SpaceWire network, however they can greatly simplify the work of the user in setting up the experiments.

### 4.1. Test Applications

In addition to the core building blocks used to build the network infrastructure, SHINe provides out-of-the-box two flexible Test Applications, one for SpaceFibre and one for SpaceWire, that can be used to study the network performances under different load conditions.

As shown in Figure 5, both the SpaceFibre and the SpaceWire Test Applications can be directly connected to the correspondent Endpoint. They can be configured by the user to simulate a generic device, in particular it is possible to set:

- Data generation rate and packet size. The test application will try to send packets of the predefined length in order to achieve the average transmission rate (Bits/sec) defined by the user;
- Data consumption rate. The test application will read the data coming from its Endpoint at a rate limited to the one specified by the user. This can be used to simulate slow destination devices that are not able to consume data at maximum speed;
- Destination nodes. A test application will try to send packets only to the nodes specified in this list. In case the path addressing is used, the chain of output ports along the path from the source to the destination is automatically derived by SHINe.

Note that in case of SpaceFibre test application, the parameters described above can be specified individually for each virtual channel. Another possible usage of these example applications is to validate the correctness of the packets content: the generated packets contain incremental integers that are checked for integrity on reception. In case of errors in the transmission (e.g., bugs in the code or error injection on links), the receiving application raises an error.

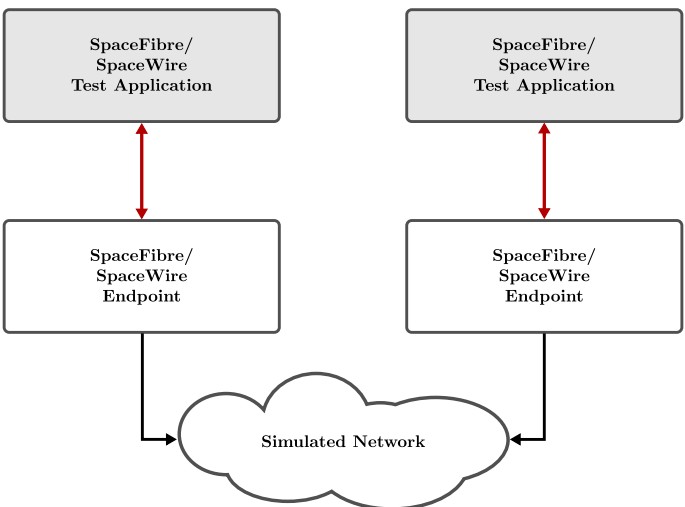

**Figure 5.** SpaceFibre and SpaceWire Test Applications.

## 4.2. RMAP Modules

Remote Memory Access Protocol (RMAP) [14] is the most common protocol used directly on top of SpaceWire. It allows accessing a remote memory region on the target device in several operation modes. The RMAP standard specifies two types of device: i) the Initiator, which is responsible for issuing the requests and ii) the Target, which must react to the requests and, if necessary, send a reply back to the Initiator. SHINe fully implements the RMAP standard specifications, for both the Initiator and the Target. They are implemented as application modules for the SpaceFibre and SpaceWire Endpoints, actually creating an additional layer between the final application of the user and the network infrastructure (see Figure 6).

The top-level user application, in order to be connected with the RMAP modules, must extend the C++ abstract classes *AppRMAPTarget* or *AppRMAPInitiator*, depending on the case. Extending these interfaces allows the underlying RMAP layer to call callback functions of the user application, for example when a new complete RMAP Request is received. Among the other features, the SHINe RMAP implementation supports:

- All the three RMAP commands: Write, Read and Read-Modify-Write;
- Both Acknowledged and non-Acknowledged commands. In the first case, an Ack is automatically sent back by the RMAP Target to the Initiator with the status of the transaction (containing an error code in case something went wrong). In the second case, the Initiator is not notified about the result of the transaction;
- Both Verified and non-Verified commands. The data field of Verified commands is covered by a Cyclic redundancy Check (CRC) code, which is checked before executing the Read or Write request;
- Multiple ongoing requests support for the RMAP Initiator. The Initiator can send multiple requests before receiving the associated replies. The pending requests are identified, associating a Transaction Identifier ID to them;

Hence, SHINe provides easy to use RMAP blocks with complete functionalities that can be used to simulate any kind of RMAP transaction for both SpaceFibre and SpaceWire.

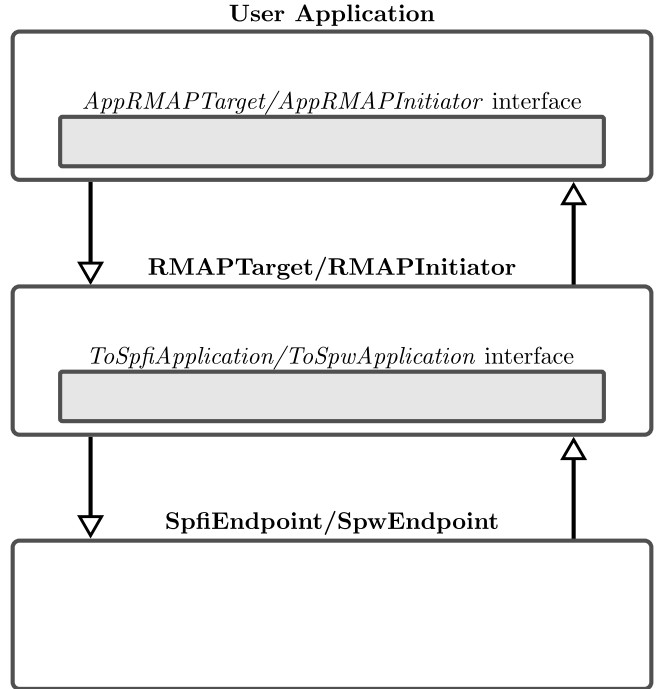

**Figure 6.** RMAP additional layer between the user application and the Endpoint.

### 4.3. Hardware-in-the-Loop

A very advanced feature of SHINe is the possibility to connect it in a Hardware-in-the-Loop (HIL) configuration with IngeniArs SpaceART [15] (see Figure 7). SpaceART [16] is a cutting-edge test equipment for the analysis of SpaceFibre and SpaceWire devices, equipped with two SpaceFibre ports, four SpaceWire ports and an Ethernet port for the connection with the Host Personal Computer (PC).

SpaceART can be used to: (i) monitor the traffic on the links, providing snapshots of the data flowing from and to the external devices; (ii) inject errors on the links; and (iii) internally produce and consume data at specific rates to simulate different load conditions. In addition to these, it allows streaming user-defined packets from the Host PC to one or more of the SpaceFibre and SpaceWire ports and vice-versa, greatly increasing the versatility of the test equipment.

Together with SpaceART, the SpaceWire PXI Analyser [17] from IngeniArs has been used in the Hardware-in-the-Loop configuration. It has similar functionality to SpaceART but it is based on the National Instruments PXI platform. As shown in Figure 8, a *HILSpfiNode* node (for SpaceFibre) and a *HILSpwNode* node (for SpaceWire) have been developed to communicate with SpaceART. They can be configured to connect one of the six hardware output ports of SpaceART with the Ethernet port. From the user perspective, the HIL nodes provide an interface equal to the correspondent Endpoint node, completely hiding the complexity of the communication with SpaceART. As a result, it is possible to plug an external SpaceWire or SpaceFibre Unit Under Test (UUT) into a fully simulated network without altering the packet stream. Some example use cases are:

- Test of an RMAP Initiator or Target UUT in a networking scenario, without the need of deploying several hardware components;
- Test of a new user-defined protocol running on top of SpaceFibre or SpaceWire. Using SHINe, it is extremely easy to analyse packets or inject errors to deeply test the UUT under different conditions.

Again, the Hardware-in-the-Loop is totally transparent for both the UUT connected to SpaceART and for the rest of the simulated nodes in SHINe. Because obviously the simulation speed is much slower than the data-rate of a real device, the communication will not happen in real-time. This means that the UUT will see a far end device that is "slow" in consuming and producing data. However, thanks to the flow control mechanisms implemented in both SpaceFibre and SpaceWire, no data will

be lost. Note that this is true as long as the UUT does not rely on any time-dependent mechanism such as timeouts, in which case the different time speed might, and probably will, affect the correct behaviour. If the UUT is not time-dependent, as it happens for raw SpaceFibre and SpaceWire data transmission, the integration in SHINe is transparent.

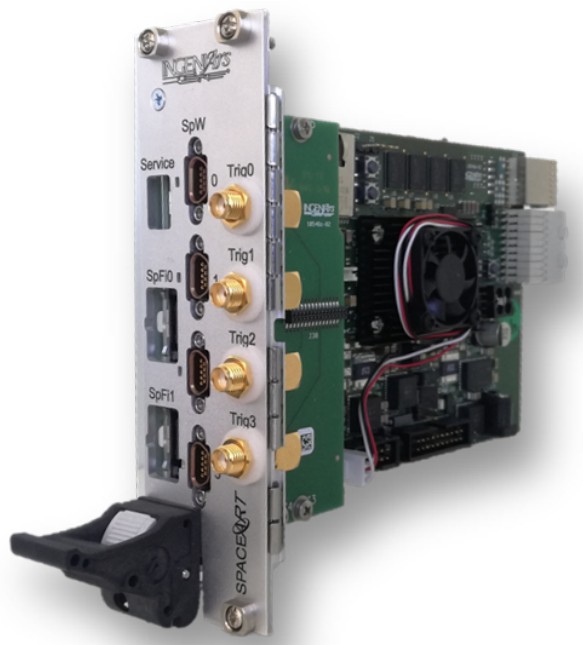

**Figure 7.** IngeniArs SpaceART.

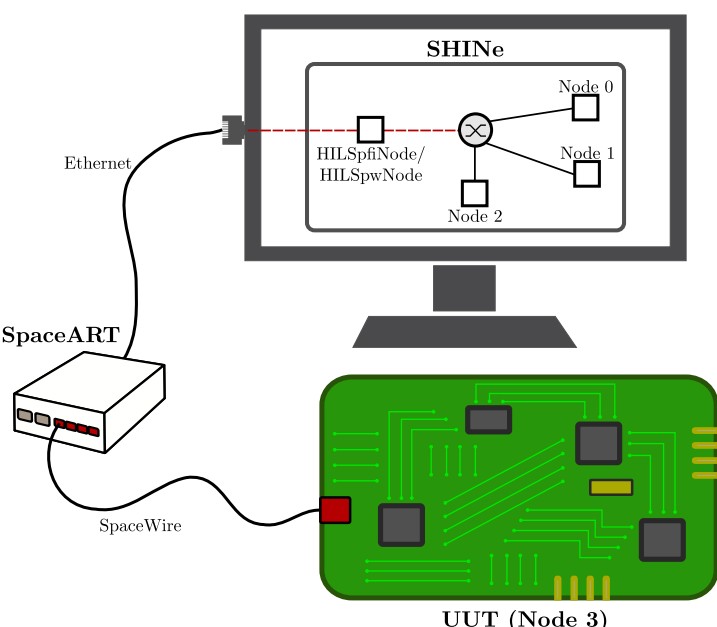

**Figure 8.** Hardware-in-the-Loop (HIL) configuration with Simulator for HIgh-speed Network (SHINe) connected with SpaceART and the Unit Under Test (UUT).

## 5. Network Setup and Results: A Case Example

In this section, an example network is set up and some of the most typical result metrics are analysed. The proposed network architecture aims to show some of the nodes illustrated in the previous sections. In particular, it comprises Test Applications, SpaceFibre Endpoints, SpaceWire

Endpoints, a Routing Switch and it makes use of the Hardware-in-the-Loop capability. The goal is to show an example of usage of SHINe from the user perspective.

### 5.1. Network Setup

The first step in the simulation of the network is its setup, intended as the deployment of the nodes composing the network itself. This step can be done either graphically, via drag&drop from a palette, or textually, using the internal language of OMNeT++ to describe network connections and parameters (called NED). In the same way, the connections between the nodes are created. As already mentioned, the network uses the Hardware-in-the-Loop capability. The SpaceWire PXI Analyser is used to simulate a Unit Under Test. It is connected to SpaceART through a SpaceWire cable, and SpaceART acts as a bridge towards the simulator. The PXI analyser can be configured to generate and/or consume data at a specific rate.

### 5.2. Nodes Configuration

After the nodes deployment, they must be configured in order to represent the scenario that the user wants to simulate. OMNeT++ allows configuring a node in two ways: (i) directly modifying the NED file of the network, either via text editor or through the OMNeT++ GUI [18] (see Figure 9) or (ii) overwriting the parameters in the *.ini* file (initialisation file).

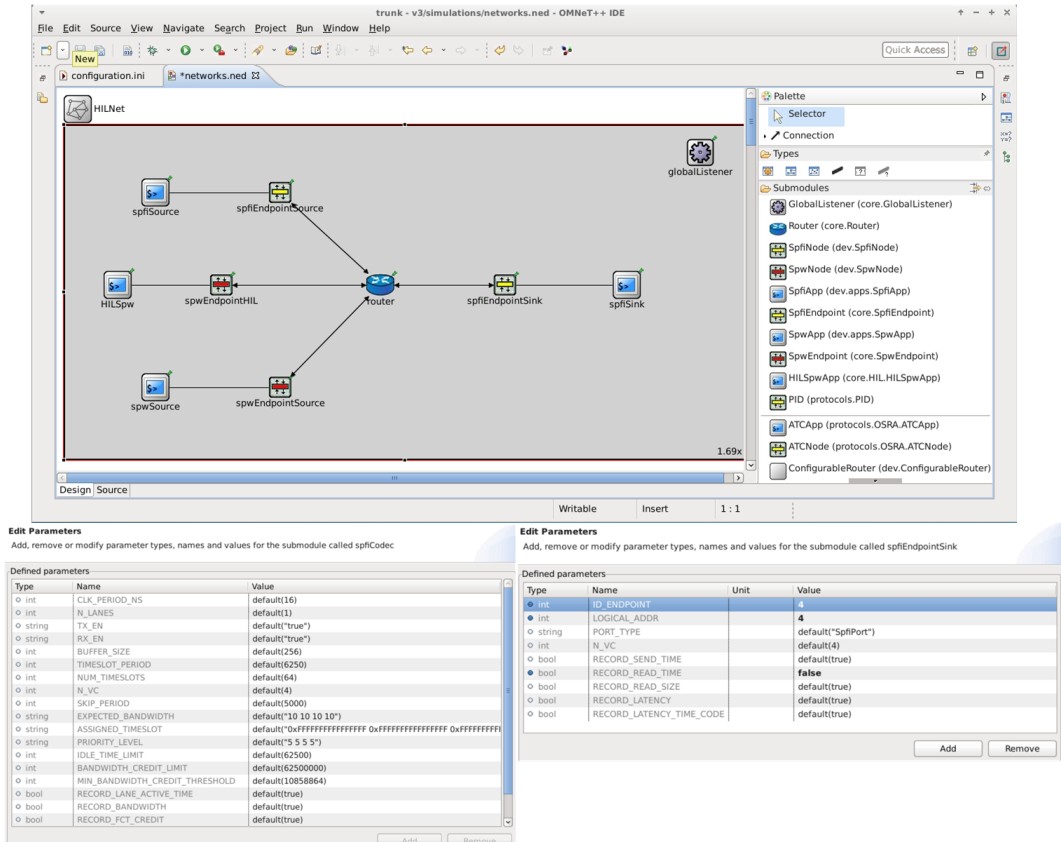

**Figure 9.** Network setup and parameter configuration for a SpaceFibre endpoint and its internal Compression/Decompression module (CODEC).

While from a practical point of view the two methods are idempotent, changing the NED file better suits "architectural" changes (e.g., the number of Virtual Channels of a node), while changing the initialisation file is preferable to configure "per-run" parameters (e.g., the Expected Bandwidth of a Virtual Channel).

In this example (see Figure 10), the network comprises the following nodes:

- a simulated SpaceFibre data sink (*spfiEndpointSink + spfiSink*) with four Virtual Channels. The application does not produce any data but consumes all incoming packets as soon as they are received;
- a simulated SpaceFibre data source (*spfiEndpointSource + spfiSource*) with four Virtual Channels. The application tries to send data on every Virtual Channel at maximum rate to the sink;
- a simulated SpaceWire data source (*spwEndpointSource + spwSource*). The application tries to send data on the link at maximum rate to the sink;
- an external hardware node (*spwEndpointHIL + HILSpw*), realising the bridge to the SpaceWire PXI analyser. The PXI analyser tries to send data at maximum rate to the sink;
- a Routing Switch, comprising two SpaceFibre ports and two SpaceWire ports.

All the SpaceFibre links are configured with a link rate of 2.5 Gbps (2 Gbps of useful data, taking into account the 8b/10b encoding), while the SpaceWire links (both simulated and physical) are set to 50 Mbps (40 Mbps of useful data, taking into account the parity and control bits).

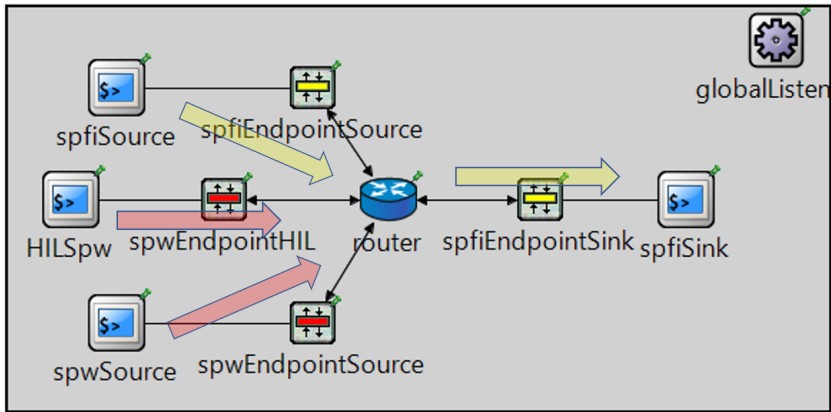

**Figure 10.** Example network simulated in SHINe.

Table 2 summarizes the main configuration parameters of the ports, with the Expected Bandwidths and the generated packets size. No protocol is being used on top of SpaceFibre and SpaceWire, so raw data are generated. The Expected Bandwidth is a SpaceFibre parameter defining the percentage of the total link bandwidth assigned to each Virtual Channel. As far as a Virtual Channel has data to send, this mechanism guarantees at least the assigned portion of the link capacity to that Virtual Channel. In case it has no data to send, this capacity is split among the other Virtual Channels. Table 3 represents the Virtual Network mapping inside the Routing Switch. Note that the SpaceWire ports are mapped to Virtual Network 1, so the wormholes will be established with Virtual Channel 1 of the output SpaceFibre port.

**Table 2.** Ports configuration parameters.

| Port | Expected Bandwidth (per VC) | Packet Length (Bytes) |
|---|---|---|
| **SpFi Data Source** | [10%, 20%, 30%, 35%] | 100,000 (all VCs) |
| **SpW Data Source** | - | 100,000 |
| **HIL Data Source** | [10%, 20%, 30%, 35%] | 100,000 (all VCs) |
| **Routing Switch (SpFi output port to Sink)** | [10%, 20%, 30%, 35%] | - |

**Table 3.** Routing Switch Virtual Networks.

| Port | Virtual Channels to Virtual Networks mapping |
|---|---|
| **SpFi Data Source Port** | [0, 1, 2, 3] |
| **SpW Data Source Port** | [1] |
| **HIL Data Source Port** | [1] |
| **SpFi Data Sink Port** | [0, 1, 2, 3] |

### 5.3. Simulation Run

Once the network is ready, the simulation can be run. OMNeT++ allows running the simulation both graphically and from command line. In case the graphical environment is chosen, different simulation speeds can be selected to trade-off execution time versus debug information printed on screen.

### 5.4. Simulation Results

During the simulation, SHINe collects several measurements to be analysed at the end of the run, either directly in OMNeT++ through the embedded data manipulation and visualization tool (*scavetool*) or exporting the data for further elaboration (e.g., in MATLAB or GNU Octave). In the following, some measurements related to the SpaceFibre output port of the Routing Switch connected to the Sink node are shown and analysed. They are:

- Virtual Channels utilization. This parameter is analysed to prove the correct implementation of the SpaceFibre protocol, especially for what concerns the Quality of Service mechanism.
- End-to-end packet latency. From this value, the impact of the wormhole routing can be studied, taking into account the interaction between the SpaceWire and SpaceFibre protocols.

For what concerns the Virtual Channel (VC) utilization in the Routing Switch output port, it is important to check the simulated results (see Figure 11) against the Expected Bandwidth values. It can be seen that VC0, VC2 and VC3 use just slightly more link capacity than expected, with the expected being 10%, 30% and 35% for VC0, VC1 and VC3 respectively, while VC1 is under-using its bandwidth. This happens because VC1 is shared between the SpaceFibre and the two SpaceWire Data Sources. When the <OutputPort, VC> of the Routing Switch is assigned to one of the SpaceWire nodes, NChars are transferred slowly, precisely at 40 Mbps (i.e., 80% of the SpaceWire input link rate). This value is much lower than the 20% of the SpaceFibre output link rate (i.e., 400 Mbps) assigned to Virtual Channel 1, causing the Virtual Channel to slowly accumulate credit. Instead, when VC1 of the SpaceFibre Data Source gets the wormhole, the accumulated credit is spent, causing a short utilization peak.

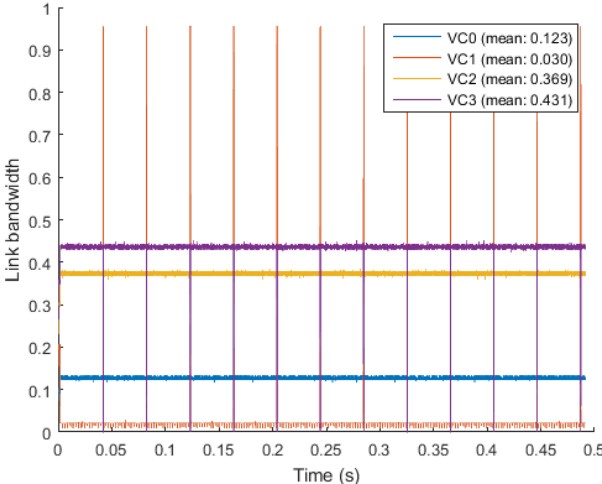

**Figure 11.** Link utilization per Virtual Channel of the Routing Switch SpaceFibre output port.

The other metric under study is the packet latency from each source to the sink. The latency is measured from the time the packet enters the transmission FIFO in the source node to the time the last byte is read out of the reception FIFO in the destination node. The results are shown in Table 4. As expected, the packets originated from the SpaceFibre source by VC0, VC2 and VC3 are not influenced by the SpaceWire nodes transmitting, being allocated to different Virtual Networks and therefore not sharing any network resource with them. Hence, the latency they experiment depends

only on the link speed and the portion of the link bandwidth (Expected Bandwidth) they have allocated (the larger the Expected Bandwidth, the faster a packet is transmitted along the link and thus the shorter its latency).

**Table 4.** Packet latency for each source node to the sink (seconds).

| Source Node/VC | Minimum | Average | Maximum |
| --- | --- | --- | --- |
| SpFi, VC0 | $3.211 \times 10^{-3}$ | $3.389 \times 10^{-3}$ | $3.701 \times 10^{-3}$ |
| SpFi, VC1 | $2.020 \times 10^{-3}$ | $34.910 \times 10^{-3}$ | $40.519 \times 10^{-3}$ |
| SpFi, VC2 | $1.086 \times 10^{-3}$ | $1.189 \times 10^{-3}$ | $1.520 \times 10^{-3}$ |
| SpFi, VC3 | $0.915 \times 10^{-3}$ | $0.997 \times 10^{-3}$ | $1.311 \times 10^{-3}$ |
| SpW Sim | $40.201 \times 10^{-3}$ | $40.688 \times 10^{-3}$ | $41.845 \times 10^{-3}$ |
| SpW HIL | $40.112 \times 10^{-3}$ | $40.392 \times 10^{-3}$ | $41.448 \times 10^{-3}$ |

The packets originated by the two SpaceWire nodes (the simulated one and the HIL) have higher latency, due to the packets being transmitted over much slower SpaceWire links. Finally, a similar latency is obtained for the SpaceFibre source node on VC1. This happens because it shares the Virtual Channel with the two SpaceWire nodes and, when they get the wormhole in the Routing Switch, the SpaceFibre packet has to wait for their slow complete transmission.

The results match exactly with the expected values, proving the correct implementation of both the SpaceFibre Quality of Service mechanism and the functionality of the Hardware-in-the-Loop.

## 6. Conclusions

In this paper, the simulator SHINe for SpaceFibre and SpaceWire networks has been presented. The main building blocks needed to set up a network have been illustrated in detail, together with some additional modules implementing advanced features like RMAP and the Hardware-in-the-Loop capability. Finally, all the steps to run a simulation of an example network have been shown, from the node deployment to the analysis of the results.

SHINe allows to quickly and easily implement and test SpaceFibre or SpaceWire-based networks and evaluate typical metrics such as packet latency and link utilization, as well as protocol-related measurements (FCT counters and buffer status). The simulator, based on the open-source framework OMNeT++, is modular and easily extensible, allowing the user to implement custom applications. SHINe can effectively help both the study of networks before their actual implementation and the development of new protocols on top of SpaceFibre and SpaceWire.

**Author Contributions:** The research and the article preparation have been carried out by A.L. and P.N. under the supervision of L.F., D.D. and D.J.

**Funding:** IngeniArs SpaceFibre technologies have been developed in the framework of the SIMPLE project (Spacefibre IMPLementation design & test Equipment). This project received funding from the European Unions Horizon 2020 research and innovation programme under Grant Agreement No. 757038.

**Conflicts of Interest:** The authors declare no conflict of interest.

## Abbreviations

The following abbreviations are used in this manuscript:

| | |
| --- | --- |
| BC | Broadcast |
| CDH | Command and Data handling |
| CRC | Cyclic Redundancy Check |
| ECSS | European Cooperation for Space Standardization |
| EGSE | Electrical Ground Segment Equipments |
| ESA | European Space Agency |

| FCT | Flow Control Token |
|---|---|
| FDIR | Fault Detection Isolation and Recovery |
| FPGA | Field Programmable Gate Array |
| In BCB | In Broadcast Channel Buffer |
| LUT | Look-Up-Tables |
| MAC | Medium Access Controller |
| Out BCB | Out Broadcast Channel Buffer |
| QoS | Quality of Service |
| Reg | Register |
| SAR | Synthetic Aperture Radars |
| VCB | Virtual Channel Buffers |
| SHINE | Simulator for HIgh-speed Networks |
| RMAP | Remote memory access protocol |
| MOST | Modelling of SpaceWire Traffic |
| SPW | SpaceWire |
| SPFI | SpaceFibre |
| FIFO | First In First Out |
| UUT | Unit Under Test |
| SUAI | Saint-Petersburg University of Areospace Instrumentation |

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
