# Peer review of "SHINe: Simulator for Satellite on-Board High-Speed Networks Featuring SpaceFibre and SpaceWire Protocols"

_aerospace, doi:10.3390/aerospace6040043_

Reviewer 1 Report

 I think that this technical paper has potential to be published to serve as a necessary tool for the spread of SpaceFibre and a satellite system network of the next generation.
We recommend that the following comments be reflected to make the content clear.

R1

In Introduction section
About L22 Single network, L28 the same network etc, it seems to point to the physical layer. It is recommended to correct the expression clearly, as the distinction between logical layer (including virtual channels) and physical layer seems unclear.

R2

In Related Works

It is recommended to put the whole in a table and show the merits and demerits.

Q1

In 3.2 SpaceFibre Endpoint 3.3SpaceWire

Is the delay time at the interface with the app defined as a configuration parameter?  For example, the part of appPacket Gate in Fig2 and Fig3.

R3

It is recommended to add the data flow to the figure in Fig. 9 of 5.2

Q2

In the Table 1, Packet Length should be understood as 100 kbytes?
What is the data transmission protocol used in the simulation? Is it RMAP?
It is recommended to add information of protocol.

Q3

Specifically, what value does the term "expected" in L51 of 5.3 refer to?

Q4

Is the L53, “the wormhole” generated because the packet size set in the simulation was large?  It is recommended to clearly specify the reason.

Q5

The relationship between the parameters shown in the Table 1 and the times shown in the Table 3 cannot be understood as “Expected bandwidth parameter” in L63. Please explain why the values in Table 3 are reasonable.

Author Response

I would like to thank on behalf of all the authors the reviewer for the precious comments. The manuscript has been revised according to the reviewer comments; please find hereafter how they have been addressed.

In Introduction section About L22 Single network, L28 the same network etc, it seems to point to the physical layer. It is recommended to correct the expression clearly, as the distinction between logical layer (including virtual channels) and physical layer seems unclear.

Clarification has been added (L28/29/30)

In Related Works It is recommended to put the whole in a table and show the merits and demerits.

Table 1 has bene added. It recaps the various protocol available in literature. However for some of them no detailed description is available in literature, as specified in section 2.

In 3.2 SpaceFibre Endpoint 3.3SpaceWire Is the delay time at the interface with the app defined as a configuration parameter?  For example, the part of appPacket Gate in Fig2 and Fig3.

By default, there is no delay between the SpaceFibre/SpaceWire Endpoint and the application. The delay in L124 and L187 refers to the delay towards the “Physical Layer” (not present here). A small change has been made to make it clear (L128).

It is recommended to add the data flow to the figure in Fig. 9 of 5.2

Added (Figure 10)

In the Table 1, Packet Length should be understood as 100 kbytes? What is the data transmission protocol used in the simulation? Is it RMAP? It is recommended to add information of protocol.

It is intended as 100kbytes, point removed for clarity. None protocol is used, raw SpaceFibre/SpaceWire data are generated (added clarification L335/336).

Specifically, what value does the term "expected" in L51 of 5.3 refer to?

Added clarification L360/361

Is the L53, “the wormhole” generated because the packet size set in the simulation was large?  It is recommended to clearly specify the reason.

The wormhole is intended as the connection established between <input port, VC> and <output port, VC>, not related to the packet length. Clarification has been added L363/364.

The relationship between the parameters shown in the Table 1 and the times shown in the Table 3 cannot be understood as “Expected bandwidth parameter” in L63. Please explain why the values in Table 3 are reasonable.

The packets originated from the SpaceFibre source by VC0, VC2 and VC3 are not influenced by the by the SpaceWire nodes transmitting, being allocated to different Virtual Network and therefore not sharing any network resource with them. Hence, the latency they experiment depends only on the link speed and the portion of the link bandwidth they have allocated. See L373-378.

I hope that the further explanation given in the paper, according to the reviewer comments, will allow the paper for being reconsidered for publication.

Once again, I would to thank you for the comments and the interesting outcomes of it that contributed to create a clearer text.

Kind regards,

Pietro Nannipieri

Reviewer 2 Report

The paper fits the journal area, presents interesting for the aerospace community R&D results and could be recommended for publication. However, Thales has presented already the MOST with the SpaceFibre simulation support also. In the paper it isn’t clear what could be watched and logged in a network simulation process, besides the final statistics that is presented in the paper.

Author Response

I would like to thank on behalf of all the authors the reviewer for the precious comments. The manuscript has been revised according to the reviewer comments; please find hereafter how they have been addressed.

Thales has presented already the MOST with the SpaceFibre simulation support also.

MOST definitely supports SpaceFibre in its OPNET version, while as March 2018, the NS3 version was not supporting it (and nothing has been presented at the SpaceWire Conference of September 2018). Clarification has been added (L to point out that the OPNET version supports SpaceFibre. Up to our knowledge we found in literature just a master’s thesis about NS3 implementation with SpaceFibre but unfortunately no access is granted to the work to general public so no information about is available.

In the paper it isn’t clear what could be watched and logged in a network simulation process, besides the final statistics that is presented in the paper.

The list of observable figures has been extended (L80/81)

I hope that the further explanation given in the paper, according to the reviewer comments, will allow the paper for being reconsidered for publication.

Once again, I would to thank you for the comments and the interesting outcomes of it that contributed to create a clearer text.

Kind regards,

Pietro Nannipieri

Reviewer 3 Report

This paper presents a new software for simulation of onboard SpaceWire and SpaceFibre networks. The presentation is clear; paper has related studies part, description of the simulation approach and examples of application. That makes it a scientific paper of a good level. Still I have some comments about it:

1) The Related studies chapter states, that "While each simulator has its own strength, SHINe is the only one supporting both SpaceFibre and SpaceWire that is free,". As long as I know, MOST already have the NS3 implementation with SpaceFibre, so the statement is not completely fair.

2) When author describe testing of a device (UUT) with a tool, they state difference between the simulation time and real time. Therefore, the speed of operation of a model would be very different. And authors say, that it would be repaired by the Flow Control. However, it seems that it would not, because there are many other problems related to the speed difference: like timeouts for the SpaceWire link, data delivery delays, different timeouts in applications and others. So it would be good to state this speed difference and to explain, what method would be useв to balance it?

3) It would be good to see the chapter describing the user interface and what particular things a user can set and tune.

4) References section should have more references, because 12 is not enough for scientific paper. I would prefer to see at least 18.

Author Response

I would like to thank on behalf of all the authors the reviewer for the precious comments. The manuscript has been revised according to the reviewer comments; please find hereafter how they have been addressed.

The Related studies chapter states, that "While each simulator has its own strength, SHINe is the only one supporting both SpaceFibre and SpaceWire that is free,". As long as I know, MOST already have the NS3 implementation with SpaceFibre, so the statement is not completely fair.

MOST definitely supports SpaceFibre in its OPNET version, while as March 2018, the NS3 version was not supporting it (and nothing has been presented at the SpaceWire Conference of September 2018). Clarification has been added (L to point out that the OPNET version supports SpaceFibre. Up to our knowledge we found in literature just a master’s thesis  (https://upcommons.upc.edu/handle/2117/127143)  about NS3 implementation with SpaceFibre but unfortunately no access is granted to the work to general public so no information about is available.

When author describe testing of a device (UUT) with a tool, they state difference between the simulation time and real time. Therefore, the speed of operation of a model would be very different. And authors say, that it would be repaired by the Flow Control. However, it seems that it would not, because there are many other problems related to the speed difference: like timeouts for the SpaceWire link, data delivery delays, different timeouts in applications and others. So it would be good to state this speed difference and to explain, what method would be used to balance it?

Correct, added clarification: Note that this is true as long as the UUT does not rely on any time-dependent mechanism such as timeouts, in which case the different time speed might, and probably will, affect the correct behaviour. If the UUT is not time-dependent, as it happens for raw SpaceFibre and SpaceWire data transmission, the integration in SHINe is transparent.

It would be good to see the chapter describing the user interface and what particular things a user can set and tune.

Added Graphical User Interface description (Figure 9  and L314)

References section should have more references, because 12 is not enough for scientific paper. I would prefer to see at least 18.

The number has been taken to 17, particularly focusing on previous work on SpaceWire and SpaceFibre endpoints.

I hope that the further explanation given in the paper, according to the reviewer comments, will allow the paper for being reconsidered for publication.

Once again, I would to thank you for the comments and the interesting outcomes of it that contributed to create a clearer text.

Kind regards,

Pietro Nannipieri